

# The effect of physical exercise on depression among college students: a systematic review and meta-analysis

Haopeng Zhang[1], Shahabuddin Bin Hashim[1], Dandan Huang[1] and Bowen Zhang[2]

[1] School of Educational Studies, Universiti Sains Malaysia, Penang, Malaysia
[2] School of Faculty of Education and Liberal Studies, City University Malaysia, Kuala Lumpur, Malaysia

## ABSTRACT

**Objective:** The goal of the present research was to evaluate the effectiveness of physical exercise intervention in enhancing psychological well-being and decreasing symptoms of depression among college students, adopting a systematic review and meta-analysis.

**Methodology:** The study was performed by searching four databases (PubMed, Embase, Web of Science, and the Cochrane Library) to determine randomized controlled trials (RCTs) exploring the impacts of physical exercise therapies among college students with symptoms of depression. The sequential execution of a meta-analyses, subgroup analyses, and publication bias analyses was accomplished utilizing the software package RevMan version 5.3.

**Results:** There were eight articles included. This research demonstrated a significant impact ($d = -0.75$, $P < 0.05$), indicating that physical exercise has a substantial impact on decreasing or mitigating depression. The subgroup analyses revealed that interventions involving physical exercise workouts lasting 12 weeks or longer ($d = -0.93$, $P < 0.05$), with physical exercise sessions lasting between 30 and 60 min ($d = -0.77$, $P < 0.05$), and with physical exercise performed minimum of three times a week ($d = -0.90$, $P < 0.05$) were the most effective in reducing symptoms of depression.

**Conclusion:** Physical exercise interventions have a beneficial impact on reducing depression among college students. The optimal mode was discovered to be college students participating in each session for a duration of 30 to 60 min, at least three times per week, and for more than 12 weeks. College students are encouraged to cultivate a consistent and long-term physical exercise routine to sustain their physical and mental health.

# INTRODUCTION

In recent decades, the incidence of mental illnesses has risen on the worldwide scene. Among the many mental illnesses, depression is a common global condition. There are around 280 million individuals worldwide who struggle with depression, and depression

Corresponding author
Shahabuddin Bin Hashim,
shah@usm.my

may become the world's number one cause of disability in 2030, as well as the disease with the heaviest economic and social burden (*GBD 2019 Mental Disorders Collaborators, 2022*; *World Health Organization, 2011*). Depression is a widespread psychological issue that can affect individuals from every aspect of life. It is defined by extended periods of sadness or diminished interest or enjoyment in things. Depression is caused by the interactions between factors of society, psychology, and biology, individuals who have undergone abuse, substantial loss, or other distressing situations are at a higher risk of getting depression. Difficulties in both academic and professional settings can also contribute to developing symptoms of depression (*World Health Organization, 2023a*).

At the end of 2019, COVID-19 initiated a worldwide health catastrophe and is regarded as a significant international public health disaster (*World Health Organization, 2023b*). This widespread occurrence not only threatens humanity but also affects psychological well-being for individuals (*Zhong, Huang & Liu, 2021*). According to a scientific brief published by the World Health Organization (WHO), there was a significant 25% increase in the global incidence of depression during the first year of the COVID-19 pandemic (*World Health Organization, 2022*). However, given the low recognition of depressive disorders in current social groups, the actual overall prevalence of depressive disorders is significantly higher than the above values (*Santomauro et al., 2021*). According to researchers in psychology and mental health specialists, the pandemic increased the number of suicides, depressions, and self-harm cases around the world as a result of the disease (*Moukaddam & Shah, 2020*). Based on data released by WHO, the annual global suicide deaths exceed 700,000 individuals (*World Health Organization, 2023c*). Simultaneously, among individuals aged 15–29, suicide stands as the fourth most prevalent cause of death (*United Nations, 2023*). Major depression disorders are more prevalent among youths compared to older individuals. The incidence major depression reaches its highest point among those aged 20–24 years and decreases as they become older (*Micah et al., 2021*). It is worth noting that college students are within this age range.

College students are individuals pursuing fundamental and professional higher education, who have either not yet completed their studies or have graduated and entered the workforce. The college students mentioned in this study refer to those who are receiving basic higher education and professional higher education and have not yet graduated. The age range of college students is generally between 18 and 22 years old, but this range is not fixed, and will be affected by factors such as personal learning, skipping and repeating grades, and the college's professional setting. The proportion of college students afflicted with psychological well-being problems including stress, anxiety, or depression has significantly increased in recent year (*Falsafi, 2016*; *Park et al., 2020*; *Pedrelli et al., 2015*). Approximately half of college students, may exhibit indications of at least one psychological well-being disorder (*Bruffaerts et al., 2018*). College students go through enormous life changes, including moving away from their families, acquiring the ability to live autonomously, meeting new friends, and adjusting to increased academic responsibilities (*Falsafi, 2016*; *Pedrelli et al., 2015*). These difficulties often arise in association with an increase in heightened levels of stress, anxiety, and depression among college students.

In the treatment of depression, traditional antidepressant medications can have side effects that may cause weight gain, sleep disturbances, and reproductive dysfunction (*National Health Service, 2021*; *Jin et al., 2011*). Physical exercise interventions are becoming increasingly promoted as an alternative therapy for depression (*Gordon et al., 2018*; *Pedersen & Saltin, 2015*; *Md Zemberi, Ismail & Leong Abdullah, 2020*; *Noetel et al., 2024*). Research has demonstrated that physical exercise therapies are equally effective in lowering depression levels (*Morres et al., 2019*; *Qaseem et al., 2016*; *Ravindran et al., 2016*). Comparably, physical exercise is easier to implement and may have a wider reach and participation (*Li et al., 2019*). Research on the association between physical exercise and depression has prompted numerous studies lately (*Chen, Zhang & Luo, 2021*; *Ormel et al., 2019*; *Cao, Zhang & Liu, 2023*; *Liu & Wang, 2024*). Multiple research have investigated the strong connection with individuals' level of physical exercise participation and their enhanced mental well-being, including a decrease in symptoms of depression (*Dishman, McDowell & Herring, 2021*; *Elbe et al., 2019*; *Schuch & Stubbs, 2019*). While, based on a study, engaging in physical exercise or having good cardiorespiratory fitness was found to have a negative association with the degree of symptoms among individuals diagnosed with major depression (*Papasavvas et al., 2016*). Physical exercise mainly includes three dimensions: intensity, frequency and duration. The American Heart Association defines intensity as the rate of energy consumption, which is an indicator of the metabolic demands of exercise. Frequency is defined as the number of exercises per day or week, usually lasting more than 10 min per session. Duration is interpreted as the time (minutes or hours) of a physical exercise session in the total amount of physical exercise time (for example, days, weeks, a year, or the past month) (*Strath et al., 2013*). Compared to previous research, there have been fewer studies investigating physical exercise therapies for treating depression among college students compared to adults. Furthermore, the previous studies provided evidence of significant and varying effects of physical exercise on depression, nevertheless, there still needs to be more clarity regarding the optimal form, intensity, duration, and frequency of physical exercise (*Fernandes, Scotti-Muzzi & Soeiro-de-Souza, 2022*; *Morres et al., 2019*; *Seshadri et al., 2020*). Although RCTs conducted in youths have demonstrated that physical exercise may enhance depressive status (*Brown et al., 2013*; *Larun et al., 2006*), the specific dosage and the correlation between engaging in physical exercise and the alleviation of symptoms associated with depression remain uncertain.

The purpose of this research was to conduct a systematic review and meta-analysis to explore the effects of physical exercise on depression and to synthesize and analyze the dose-effect of different physical exercises protocols among college students subjects on depression.

## MATERIALS AND METHODS

### Protocol and registration

The protocol for this systematic review was registered on February 19, 2024 in the International Prospective Register of Systematic Reviews with the PROSPERO-ID CRD42024514264.

### Literature search strategy

The study was performed by the standards outlined in the Cochrane Handbook for the Systematic Review of Interventions (*Chandler et al., 2019*) and the PRISMA Statement Specification for Systematic Review and Meta-analysis (*Moher et al., 2015*).

In February 2024, we performed a systematic literature search in the PubMed, Embase, Cochrane Library, and Web of Science databases. The search methodology employed in each database involved utilizing a combination of distinct medical subject headings (MeSH) or synonyms with the goal of discovering and evaluating pertinent studies (Supplemental Material Table A1). The search phrases from each category were merged to identify all pertinent literature databases.

## CRITERIA FOR INCLUSION AND EXCLUSION

### Selection procedure

To identify whether the meta-analysis was suitable for this article, the individual studies that were included had to meet the eligibility criteria shown in Table 1.

Data was imported from the pertinent literature into Endnote (version X9) for grouping. Subsequently, two authors (HZ and DH) independently screened duplicate results. The screening procedure entailed the evaluation of titles, review articles, and conference papers. Subsequently, the abstracts were reviewed to exclude studies that do not fulfill particular criteria, including the study subjects or the interventions used.

Finally, the full text of selected article were thoroughly checked for completeness to remove those that were not usable, were not written in English, and did not provide the data of endpoint measures.

The procedure entails a preliminary assessment of suitable articles, a deliberation on any inconsistencies, and the establishment of a consensus with the author (SH). In the end, a total of eight article were identified as suitable for this study. The Preferred Reporting Items for Systematic Reviews and Meta-Analyses flowchart provides comprehensive details on these phases, as shown in Fig. 1.

### Data extraction and quality evaluation

Two authors (HZ and DH) utilized an existing data extraction form to extract and document the succeeding data: (1) Essential article details include the identity of the first author's name, the geographical area where the study was conducted, and the year of publication; (2) Essential details include the individual's age, the target sample, and the number of participants in the study; (3) The number of participants in the intervention group and the control group, the intervention mode of the intervention group and the requirements of the control group; (4) Physical exercise variables include factors such as the type of exercise, the cycle of exercise, the frequency of exercise, and the duration of exercise. The information on the general characteristics of the eight meta-analyses selected for inclusion was extracted by two reviewers who used the data extraction form and summarized in Table 2.

The Cochrane 5.1 handbook was used to assess the quality of bias of the selected articles. The evaluation criteria for this research are as follows: The investigation assessed whether
**Table 1 Eligibility criteria by category (PICOS).**

| Category | Eligibility criteria |
|---|---|
| Population | The college students mentioned in this study refer to those who are receiving basic higher education and professional higher education and have not yet graduated. The age range of college students is generally between 18 and 22 years old, but this range is not fixed, and will be affected by factors such as personal learning, skipping and repeating grades, and the college's professional setting. |
| Intervention | Intervention consisting of physical exercise or physical activity, such as Biodanza, high intensity yoga (HIY), High-Intensity Interval Training (HIIT), pilates, Tai Chi Chuan, Baduanjin, and resistance training. |
| Comparator | Control group |
| Outcome | Benefits of physical exercise Effect size reported |
| Study design | Randomized controlled trials |

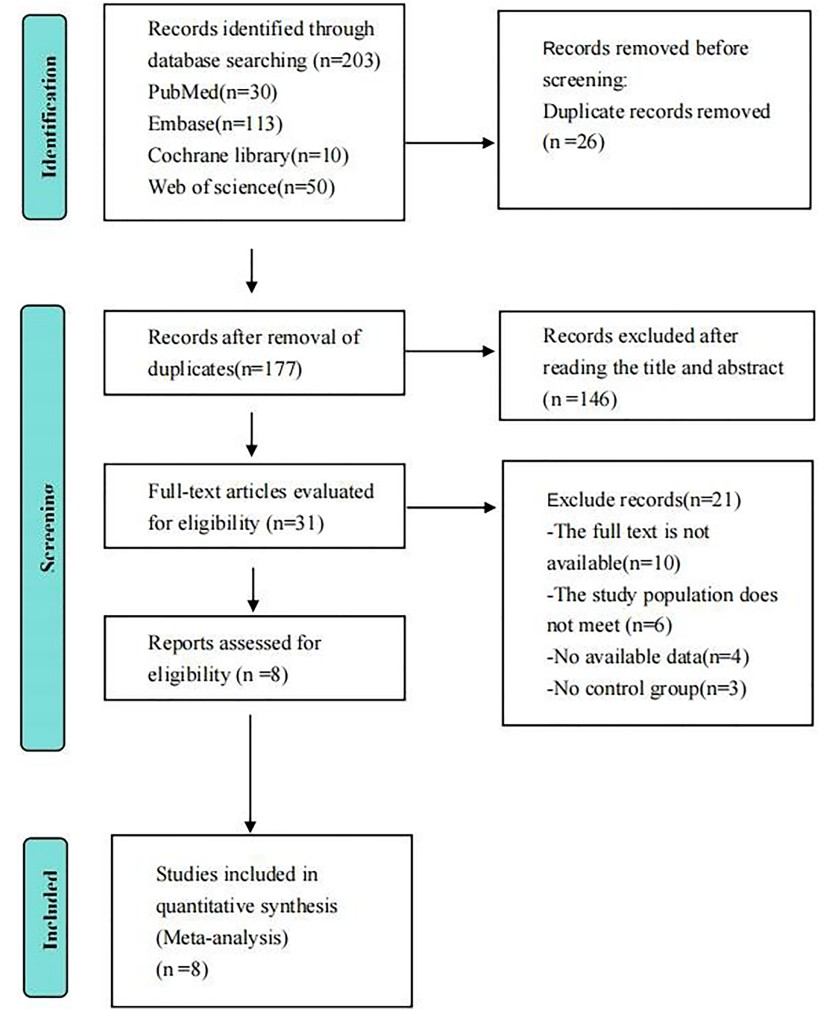

**Figure 1 PRISMA diagram depicting the sequential steps of the selection process.**

**Table 2  Summary of features of included intervention.**

| First author year | Region | Study design | Target sample | Age | Sample size (T/C) | Intervention description | Depression measurement | Intervention cycle | Duration | Frequency |
|---|---|---|---|---|---|---|---|---|---|---|
| *López-Rodríguez et al. (2017)* | Spain | RCT | University students | 22.33 ± 4.12 | Intervention: n = 42; Control: n = 53 | Biodanza | CES-D | 4 weeks | 90 min | 1 session/ week |
| *Papp et al. (2019)* | Sweden | RCT pilot | Students | Median age: 25 years | Intervention: n = 21; Control: n = 23 | High intensity yoga (HIY) | HADS | 6 weeks | 60 min | 1 session/ week |
| *Philippot et al. (2022)* | Belgium | RCT | University students | The control group 20.93 ± 1.94; HIIT groups 20.69 ± 1.44 | Intervention: n = 11; Control: n = 14 | High-Intensity Interval Training | DASS-21 | 4 weeks | 10 min | 3 sessions/ week |
| *Saltan & Ankaralı (2021)* | Turkey | RCT | University students | Pilates18.82 ± 1.071; therapeutic exercise program18.85 ± 2.495; Control group 19.42 ± 1.378 | Pilates: n = 29; Therapeutic exercise program: n = 28; Control: n = 35 | Pilates exercise; Therapeutic exercise | BDI | 12 weeks | 40–60 min | 3 sessions/ week |
| *Zhang et al. (2018)* | China | RCT | College students | 18.41 ± 2.01 | Intervention: n = 32; Control: n = 30 | Mindfulness-based Tai Chi Chuan | PHQ-9 | 8 weeks | 90 min | 2 sessions/ week |
| *Zhang & Jiang (2023)* | China | RCT | College students | Average age of 19.2 | Intervention: n = 34; Control: n = 39 | Baduanjin exercises | SCL90 | 12 weeks | 60 min | 3 sessions/ week |
| *Zhang et al. (2023)* | China | RCT pilot | College students | BWTC group 24.20 ± 4.07; Control group 22.50 ± 5.95 | Intervention: n = 9 ; Control: n = 9 | Bafa Wubu of Tai Chi | SDS | 8 weeks | 60 min | 5 sessions/ week |
| *Zhao et al. (2023)* | China | RCT | College students | 21.20 ± 2.10 | AE group: n = 29; RT group: n = 29; Control group: n = 28 | Aerobic exercise; resistance training | SDS | 12 weeks | 40–60 min | 3 sessions/ week |

the randomly chosen allocation process and the secrecy of the allocating plan were intentionally obscured or not, the blinding of participants and assessment of outcomes, as well as the completeness of outcome data, are being considered, selective report findings, and other bias. Each criterion is evaluated based on its level of risk, which can be classified as low (indicating that the criterion is met), high (indicating that the criterion is not satisfied), or medium (if not indicated). A comment is included to explain the rationale behind each assessment.

## Statistical analysis

This study employed a statistical software, Review Manager 5.3, to amalgamate effect sizes and evaluate bias. The original literature included in this study did not achieve scale consistency in the measurement of depression indicators, therefore, in order to assess effect
sizes more accurately, all data had to be converted uniformly using standardized mean difference (SMD) and selecting 95% confidence intervals (CI). The SMD is calculated as the discrepancy between the means of the pre-and post-intervention measurements, divided by the final combined standard deviation (SD) value. This approach overcomes the issue of inconsistent measuring units across multiple scales.

This study employed the Cochrane Q-test to ascertain the level of heterogeneity based on the $I^2$ value. If the measured $I^2$ value is less than or equal to 50% and $P > 0.1$, it indicates that there is no substantial heterogeneity present in this study. Ultimately, this study employed a random effects model to assess publication bias by employing funnel plots and to assess the reliability of the results.

## RESULTS

### Search results

Figure 1 demonstrates that a thorough search yielded a total of 203 articles by searching PubMed ($n = 30$), Web of Science ($n = 50$), Embase ($n = 113$), and The Cochrane Library ($n = 10$). Following the process of deduplication, a total of 177 articles were acquired. Following an initial screening process, a total of 31 articles were acquired. After conducting a thorough review of the articles, including reading the complete texts and rejecting publications that did not meet the criteria for a randomized controlled trial (RCT), such as those with inadequate study design, intervention/control groups, research purpose, or outcome measures, as well as articles with inaccessible data, a total of eight articles were selected for the research.

### Basic characteristics of the articles

The meta-analysis of this study included a total of eight articles, which included a total of ten studies. The overall sample size consists of 495 participants, the intervention group contained 264 individuals, whereas the control group contained 231 individuals. The intervention cycles ranges from 4 to 12 weeks. The duration of the intervention varies from 10 to 90 min, and the frequency varies between 1 to 5 sessions each week. The interventional therapies mostly targeted aerobic exercises and resistance training.

### Evaluation of quality

The presence of design, conduct, analysis, and reporting flaws in randomized trials might hinder the ability to make accurate causal conclusions, resulting in either an underestimate or an overestimate of the actual intervention bias (*Wood et al., 2008*). Nevertheless, determining the precise impact of biases on the outcomes of a specific trial is typically unattainable (*Higgins et al., 2011*).

The primary purpose of utilizing the Cochrane Risk of Bias Tool is to evaluate the methodological rigour and potential bias in medical research, specifically in randomized controlled trials (RCTs). The Cochrane Collaboration established a method to assist researchers, clinicians, and policymakers in identifying the potential for bias that could impact the dependability of research results. The Cochrane Risk of Bias Tool includes a
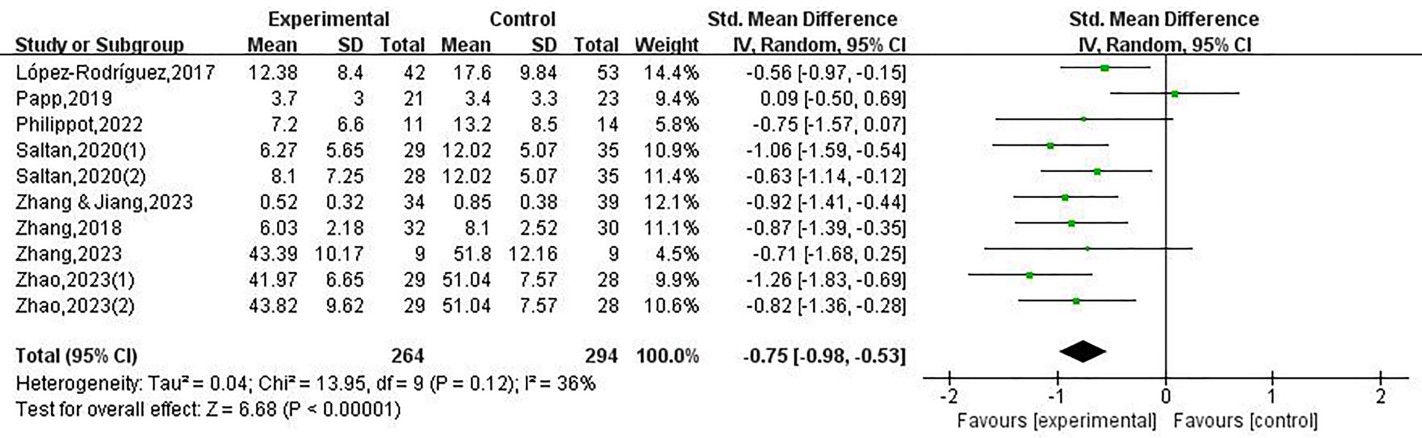

**Figure 2** Forest plot illustrating the impact of physical exercise on depression among college students (*López-Rodríguez et al., 2017*; *Papp et al., 2019*; *Philippot et al., 2022*; *Saltan & Ankaralı, 2021*; *Zhang & Jiang, 2023*; *Zhang et al., 2018*, *2023*).

total of six of bias: selection bias, performance bias, detection bias, attrition bias, reporting bias, and other biases (*Chandler et al., 2019*).

This study analyses the existing literature on the random assignment method, with a specific emphasis on seven studies that fulfill the criteria for inclusion (*López-Rodríguez et al., 2017*; *Philippot et al., 2022*; *Saltan & Ankaralı, 2021*; *Zhang et al., 2023*, *2018*; *Zhang & Jiang, 2023*; *Zhao et al., 2023*). The remaining research study does not provide specific information regarding the randomization technique (*Papp et al., 2019*). Only two articles explicitly informed readers that their study used a hidden allocation scheme (*Zhang & Jiang, 2023*; *Zhao et al., 2023*), while the remaining six articles did not mention whether the allocation scheme was hidden or not. Due to the nature of this study focusing on exercise intervention, blinding of participants may not be feasible, consequently, the participants were not blinded. Thus, none of the eight articles were considered low risk. All of the research in the eight articles demonstrated no instances of subject or data loss, and were deemed to have a low risk level. Each of the investigations included in the analysis was found to be devoid of any additional selective reporting or prejudice and was considered to have a negligible risk of bias.

## Meta-analysis results

Forest maps were utilized to conduct heterogeneity testing. The findings indicated a moderate level of heterogeneity across the research investigations ($I^2 = 36\%$, $P = 0.12$). The test for the combined effect size presented a significant statistical result (SMD = −0.75, 95% CI [−0.98 to −0.53], Z = 6.68, $P < 0.001$)as depicted in Fig. 2.

## Tests for bias

As depicted in Fig. 3, the study utilized endpoint markers for assessment, as well as the funnel plot displayed a symmetrical shape, showing the absence of major publication bias.

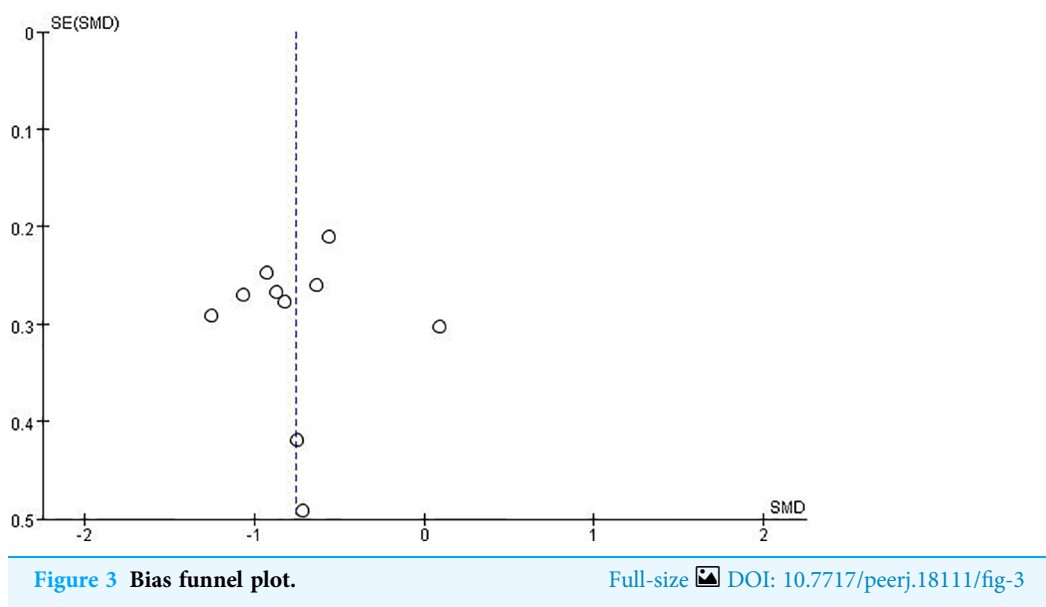

**Figure 3  Bias funnel plot.**

## Impact of physical exercise on depression among college students

Statistical tests for heterogeneity were conducted on the publications that were incorporated into the analysis. Out of these, the remaining nine studies (containing seven articles) demonstrated that physical exercise decreases depressed states among college students, except for one study. The scholars utilized a random effects model for data collection on the outcome indicators of the research. This research adopted a total of ten studies including an overall of 495 participants, comprising 264 participants randomized to the intervention group with 231 participants randomized to the control group. This study presents empirical evidence to substantiate the efficacy of implementing a physical exercise intervention in mitigating the deleterious effects of depression symptoms among college students (SMD = −0.75, 95% CI [−0.98 to −0.53], Z = 6.68, $P < 0.05$), as depicted in Fig. 2.

## Analyses on subgroups

The meta-analysis of a physical exercise intervention on depression among college students revealed a substantial level of heterogeneity in the combined effect size data. The achievement was attained through the analysis of subgroups, considering intervention cycle, duration and frequency as possible affecting impacts. The outputs of subgroup studies investigate impacts on the intervention cycle, duration, and frequency.

Regarding the intervention cycle, the studies were categorized into three distinct groups for analysis: less than or equal to 4 weeks (included two studies), between 4 and 8 weeks (included three studies), more than or equal to 12 weeks (included five studies). The study found that participating in physical exercise has been proven to alleviate depression among college students. Specifically, intervention cycles lasting 4 weeks or less (SMD = −0.60, 95% CI [−0.97 to −0.23], $P < 0.05$), as well as intervention cycles lasting 12 weeks or more had an SMD of −0.93 (95% CI [−1.16 to −0.69], $P < 0.05$), were effective in decreasing depressive symptom among college students. Conversely, the intervention cycle lasted

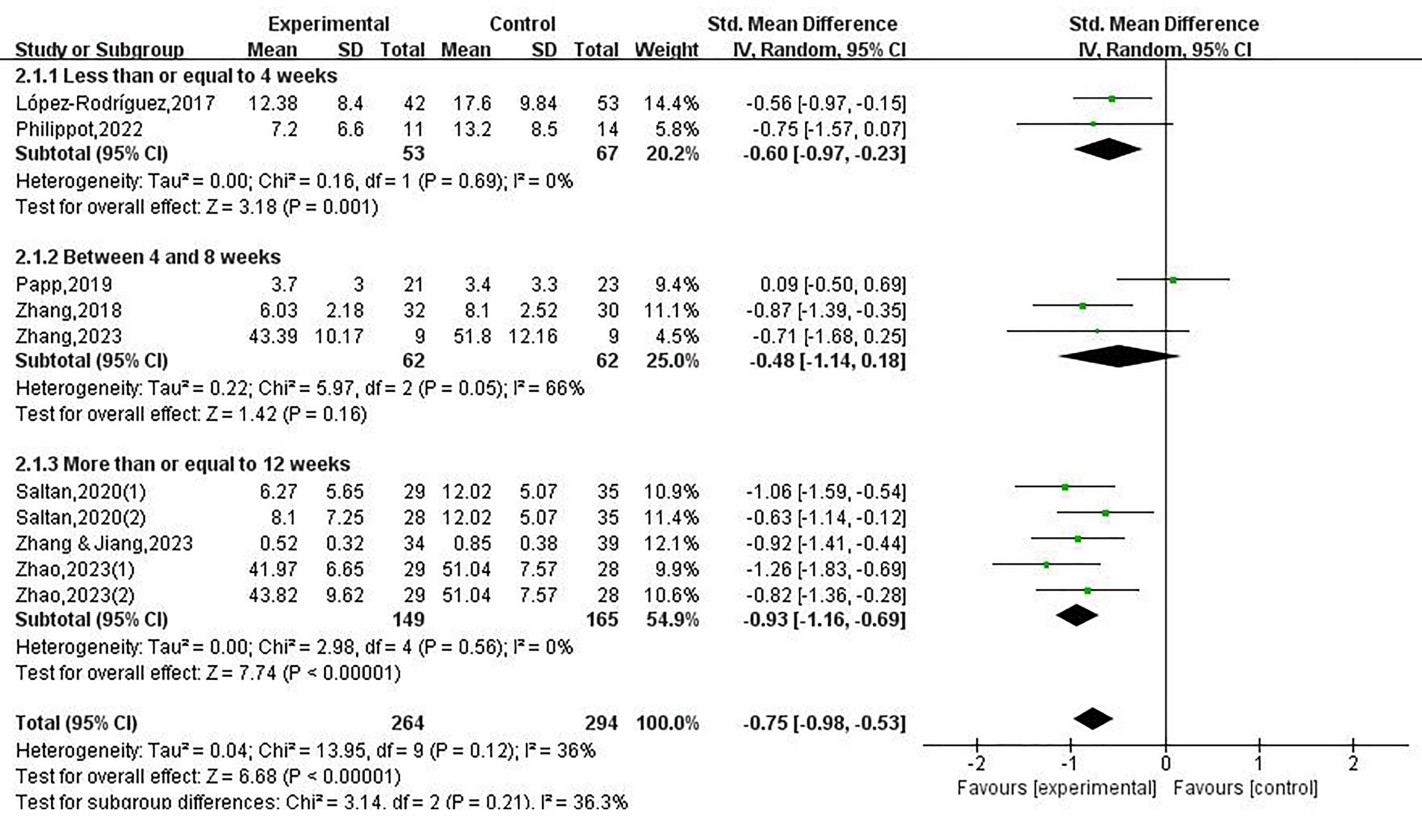

**Figure 4 Forest plot illustrating the impact of physical exercise on depression among college students within different subgroups of the intervention cycle** (*López-Rodríguez et al., 2017*; *Papp et al., 2019*; *Philippot et al., 2022*; *Saltan & Ankaralı, 2021*; *Zhang & Jiang, 2023*; *Zhang et al., 2018*, *2023*).

between 4 and 8 weeks ($P > 0.05$), there was no notable decline in depressive symptoms observed among college students.

Regarding the duration, the studies were categorized into three distinct groups for analysis: less than or equal to 30 min per session (included one study), the duration of each exercise intervention varies between 30 and 60 min, as reported in seven investigations. Additionally, two studies contained sessions that lasted more than 60 min. The research on the advantageous impacts of exercise on depressive revealed that intervention program enduring 30 to 60 min (SMD = −0.77, 95% CI [−1.09 to −0.45], $P < 0.05$), as well as sessions lasting more than 60 min (SMD = −0.68, 95% CI [−1.00 to −0.36], $P < 0.05$), were effective in decreasing depressive symptom among college students. In contrast, the intervention duration lasted for 30 min or less per session ($P > 0.05$), there was no notable decline in depressive symptoms observed among college students.

Regarding the frequency, studies were categorized into two distinct groups for analysis: less than 3 sessions per week (included three studies), and more than or equal 3 sessions per week (included seven studies). This research on the influence of physical exercise on depression showed that participating in physical exercise at a frequency of three or more sessions per week led to a substantial decrease in depression among college students (SMD = −0.90, 95% CI [−1.12 to −0.68], $P < 0.05$). On the other hand, when the

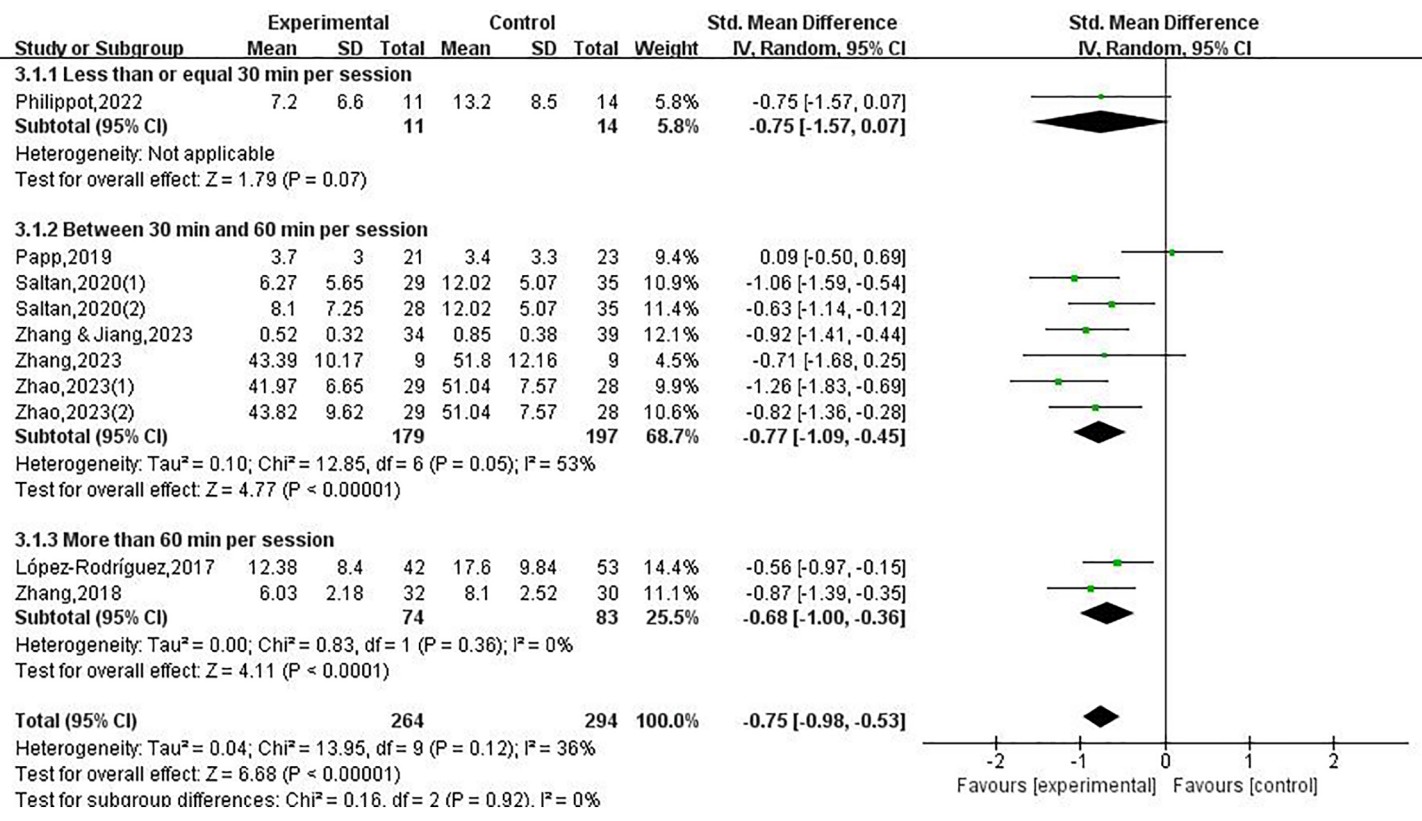

**Figure 5** Forest plot illustrating the impact of physical exercise on depression among college students within different subgroups of the intervention duration (*López-Rodríguez et al., 2017*; *Papp et al., 2019*; *Philippot et al., 2022*; *Saltan & Ankaralı, 2021*; *Zhang & Jiang, 2023*; *Zhang et al., 2018, 2023*).

intervention frequency was fewer than 3 sessions per week ($P > 0.05$), there was no notable decline in depressive symptoms observed among college students. Therefore, the intervention cycle, duration, and frequency are key factors that contribute to the observed difference in depression, as demonstrated in Figs. 4 to 6.

## DISCUSSION

After reviewing previous studies, it was found that there are currently few meta-analysis studies that have specifically targeted improving depression among college students by examining different types, cycles, frequencies, and duration of physical exercise. The research utilized a systematic review and meta-analysis to assess the impact of physical exercise intervention on depression among college students. The aim was to synthesize existing research and evaluate the magnitude of the effect of the intervention.

This current research attempts to examine meta-analyses that specifically investigate the effect of physical exercise on indicators of depression among college students. Based on *Cohen*'s *(1992)* criterion, the impacts were divided into three groups following the guideline criteria: effect sizes can be categorized as small (d = 0.20), medium (d = 0.2–0.50), or large (d ≥ 0.80). The overall effect size analysis of the selected meta-analyses provided a medium to large impact size (d = −0.75) for physical exercise in reducing depression symptoms. Similar results were found in adults, perinatal women,

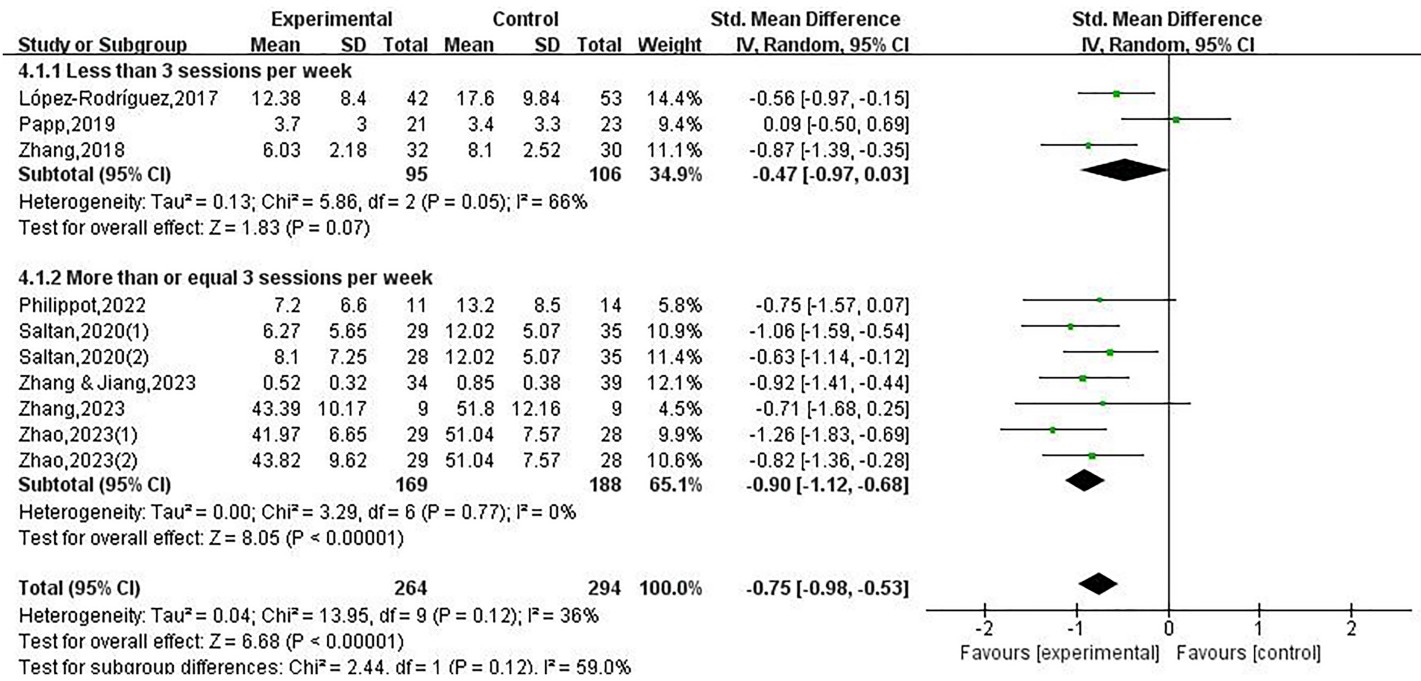

**Figure 6 Forest plot illustrating the impact of physical exercise on depression among college students within different subgroups of the intervention frequency (*López-Rodríguez et al., 2017*; *Papp et al., 2019*; *Philippot et al., 2022*; *Saltan & Ankaralı, 2021*; *Zhang & Jiang, 2023*; *Zhang et al., 2018, 2023*).**

children, and adolescents. In all these groups, the effect sizes indicating the impact of the intervention ranged from small to moderate (d = −0.48) (*Wegner et al., 2020*), and large (d = −1.02) (*Wang, Tian & Luo, 2023*).

The current study demonstrated an effect with a value d = −0.75, the effect size is medium to large range (*Cohen, 1992*). Depressed individuals also showed similar results (d = −0.76), indicating the positive effects of physical exercise (*Correia et al., 2023*). Further investigation is required to encompass clinical cohorts of college students, as there is a dearth of data about this particular demographic. Based on these findings, it may be postulated that physical exercise may be a pertinent intervention for depressive symptom in both college students and adults. These research findings confirm prior studies that show college students can gain advantages from engaging in physical exercise and experience notable enhancements in their depression levels.

Numerous examples of related research reviews demonstrate that physical exercise may prove equally beneficial as psychotherapy and medicine in alleviating mild to moderate depressive symptoms (*Cooney et al., 2013*; *Danielsson et al., 2014*). Previous research has proven that physical exercise may function as a substitute.to antidepressant medication for reducing depression (*Guerrera et al., 2020*; *Hidalgo et al., 2019*). Furthermore, physical exercise has been suggested as a primary therapeutic approach to individuals with mild to moderate depression (*Rethorst, Wipfli & Landers, 2009*).

The present study demonstrated that physical exercise, such as Biodanza, high intensity yoga (HIY), High-Intensity Interval Training (HIIT), pilates, Tai Chi Chuan, Baduanjin, and resistance training, had a significant effect in reducing or preventing depression

among college students. It is worth noting that a common element across these studies was the incorporation of aerobic exercises. Research has demonstrated that engaging in aerobic exercise yields beneficial outcomes for both physical and mental, effectively reducing symptoms of depression (*Choo et al., 2014*).

The rate of enhancement of depression symptoms among college students is intricately linked to the cycle, duration, and frequency of physical exercise. The intervention cycle of physical exercise is highly varied and requires confirmation to determine the optimal duration. This research shown that a period exceeding 12 weeks had a significant impact on reducing depression levels among college students, aligning with *a prior* investigation (*Carter et al., 2019*). This finding aligns with a previous study that suggested college students should participate in physical exercise sessions lasting between 30 to 60 min, at least three times per week (*Dipietro et al., 2019*).

Nevertheless, although physical exercise performs an essential function in alleviating depression status among college students, the optimal form of physical exercise needs to be further explored, as the best form of physical exercise still remains to be evidenced (*Dipietro et al., 2019*).

## CONCLUSION

The research employed meta-analysis for analyzing the importance of physical exercise as an intervention for depressive symptom among college students. Intervention cycle, duration and frequency may be the main factors affecting the study results. The present study suggests integrating suitable physical exercise into the routines of individuals experiencing depressive symptoms as a means to substantially alleviate depression and enhance both physical and mental well-being. After comparing the different intervention cycle, duration and frequency, it is recommended that physical exercise for college students engage in each session between 30 to 60 min, more than or equal 3 sessions per week, and physical exercise sessions last longer than 12 weeks to develop a long-term habit of regular physical exercise. The methodology aims to reduce depression among college students, thus facilitating optimal outcomes.

This study still has several shortages. Firstly, the study participants consisted of college students. Hence, there might be constraints in generalizing the results to individuals within the same age bracket, such as employed young adults or women in their reproductive years. Furthermore, the restricted number of incorporated studies may have resulted in a certain level of selection bias, and the small sample sizes can impact the accurate outcomes of subgroup analysis. At last, a total of seven depression scales were used throughout the ten studies included in this study. The various number of items examined in each scale may have led to variations in detection rates, thus impacting the study's findings. Furthermore, the 10 studies did not include any information regarding the intensity of physical exercise, including heart rate, oxygen uptake, and respiratory rate. Hence, further research could focus on filling this research gap.

### Funding
The authors received no funding for this work.

### Competing Interests
The authors declare that they have no competing interests.

### Author Contributions
- Haopeng Zhang conceived and designed the experiments, performed the experiments, analyzed the data, prepared figures and/or tables, authored or reviewed drafts of the article, and approved the final draft.
- Shahabuddin Bin Hashim conceived and designed the experiments, performed the experiments, analyzed the data, authored or reviewed drafts of the article, and approved the final draft.
- Dandan Huang conceived and designed the experiments, performed the experiments, analyzed the data, prepared figures and/or tables, and approved the final draft.
- Bowen Zhang performed the experiments, prepared figures and/or tables, and approved the final draft.

### Data Availability
This is a Systematic review and Meta-analysis.

### Supplemental Information
Supplemental information for this article can be found online at http://dx.doi.org/10.7717/peerj.18111#supplemental-information.

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
