# Peer review of "The effect of physical exercise on depression among college students: a systematic review and meta-analysis"

_PeerJ, doi:10.7717/peerj.18111_

## Round 0.1 · original submission · Major Revisions

Please attend to all of the key points raised by all 3 reviewers - all are experts in this area of research. Reviewer 1 has raised important issues about the basic reporting of meta-analyses and also whether the searches were comprehensive. Reviewer 3 has provided detailed comments on a copy of the script (see the attached document).

I would also like a revision to attend to the study numbers in the PRISMA flow chart - as they do not seem to make sense to me as currently recorded. Could I also please ask you to check and confirm the Cochrane RoB2 assessments - especially for attrition bias and reporting bias, which both suggest all studies are at low risk of bias - the latter (reporting bias) would seem to depend largely upon studies being preregistered and it doesn't seem as if they are all preregistered.

Reviewer 1 ·

Basic reporting

There are some general errors throughout the document especially with spaces after full stops. Sentences are quite often short which makes it difficult to follow, especially in the introduction. More citations are needed to support some statements about factual information in particular around depression.

Experimental design

The research question comes as a bit of a surprise at the end of the intro. It includes duration which has not been discussed prior to the question. I would therefore go back and include it within the intro.

The experimental design is not clear as there is no inclusion and exclusion criteria. Discussion around the fact that there are different measures of depression is needed some where. A lot more detail needs to go in to the methods and understanding why articles were selected.

Validity of the findings

There are only papers included from 2017 – I am sure there will have been more papers prior to this. There are only 8 included but I am sure there should be more, but without seeing the inclusion and exclusion criteria it is very difficult to comment. I don't feel you can answer your research question from the limited experimental design that I can see.

Additional comments

There is a lot of repetitiveness in the intro with physical exercise. The flow needs to be improved as well as correcting the spaces when there is a full stop etc. There are lots of short sentences which often don’t cite a reference where needed. The research question includes exercise duration which was not mentioned in the intro, so not sure where this has come from. Some text needs to be included around duration in the intro. There are a lot of references around COVID but we should know some of the impact now rather than stating that it will affect.

The protocol was published on the 19th Feb but you closed the search for publications on the 21st? was there only 2 days of searching? It is not clear what your inclusion and exclusion criteria are? You only found approximately 200 studies which I find surprising, I would think there would be more. I am not certain how you came to your 8 to be included. The methods needs to be written in past tense – some sections aren’t. It is not clear at the end of the methods what papers are being selected and why they are being selected and what outcome variables are being considered. There are different measures for depression, but which one are you focusing on?

In the discussion you mention that you examined meta-analyses which you didn’t do.

Reviewer 2 ·

Basic reporting

The main research content of this paper is a systematic review and meta-analysis of the impact of physical exercise on depressive symptoms in college students. The literature background section elaborates in detail on the prevalence and severity of global depression, as well as the impact of the COVID-19 pandemic on mental health, thereby providing ample practical significance and urgency for the research question. The research question is clearly defined, pointing out the knowledge gap that the study aims to fill and explaining the importance of the research. However, there are some literatures that the paper has not covered. The following are some literatures related to physical exercise that should be included in the author's literature introduction section, and their results and conclusions should be discussed in the discussion section.

Cross-Lagged Relationship between Physical Activity Time, Openness and Depression Symptoms among Adolescents: Evidence from China. International Journal of Mental Health Promotion, 2023, 25, 1009–1018. https://doi.org/10.32604/ijmhp.2023.029365

Unlocking the Power of Physical Activity in Easing Psychological Distress. World Journal of Psychiatry, 2024, 14, 1–7. https://doi.org/10.5498/wjp.v14.i1.1

Experimental design

This paper, through systematic review and meta-analysis methods, comprehensively assesses the impact of physical exercise on depressive symptoms in college students. The study design is rigorous, employing extensive database searches, focusing on randomized controlled trials, and utilizing appropriate statistical methods to combine effect sizes and evaluate heterogeneity. The study also includes risk of bias assessment and publication bias detection, enhancing the credibility of the results.

Validity of the findings

While the results indicate that physical exercise has a significant impact on depressive symptoms among college students, to enhance the generalizability and reliability of the conclusions, future research should consider expanding the sample size, using consistent measurement tools, and further exploring the best intervention models for different types of physical exercise.

Additional comments

no comment

·

Basic reporting

There are quite a few errors in expression and a lack of clarity in places. Some references also need to be updated.

The article is structured in accordance with systematic reviews and meta-analyses.

All the tables and figures are appropriate.

Experimental design

The design was fine and the study answers an important question on the link between physical activity and depression in young people.

The methods were mostly sufficient for replication.

Validity of the findings

The findings are valid as the methodological processes followed were all appropriate.

Additional comments

See all my highlighted points in the attached paper.

---

## Round 0.2 · accepted · Accept

Thank-you for addressing the points made by the three reviewers and my own suggestions. Having examined the manuscript, I am happy to suggest acceptance, and the paper is now ready for publication.

Reviewer 2 ·

Basic reporting

good

Experimental design

good

Validity of the findings

good

·

Basic reporting

The authors have made all the suggested changes and clarifications I suggested in my original review. The article has been significantly improved.

Experimental design

All suggested changes have been made.

Validity of the findings

All suggested changes have been made.